



# Soil Moisture Estimation Based on Probabilistic Inversion over Heterogeneous Vegetated Fields Using Airborne PLMR Brightness Temperature

Chunfeng Ma[1], Xin Li[1*, 2, 3], Shuguo Wang[4,1]

*Corresponding author: Xin Li (lixin@lzb.ac.cn)

[1]Key Laboratory of Remote Sensing of Gansu Province, Northwest Institute of Eco-Environment and Resources, Heihe Remote Sensing Experimental Research Station, Chinese Academy of Sciences, Lanzhou, 730000, China

[2]CAS Center for Excellence in Tibetan Plateau Earth Sciences, Beijing, 100101, China

[3] University of Chinese Academy of Sciences, Beijing 100049, China

[4]Jiangsu Normal University, No. 101 Shanghai Road, Xuzhou, P. R. China

## Abstract

The L-band radiometer has demonstrated its strong ability of estimating soil moisture (SM) over vegetated surface. However, the present SM products derived from satellite radiometers are hardly directly used in heterogeneous oasis. This may be attributed to the coarse spatial resolution of the satellite-based observations and the defects of the point-estimation algorithms which cannot quantify the uncertainty in SM inversion. This paper presents a SM estimation based on the combination of Bayesian probabilistic inversion with high resolution airborne radiometer observations. The overall objective is to quantify the uncertainty in SM estimation and to provide a desirable SM estimation. Two retrieval strategies (2P and 3P strategy) are performed based on 6-channel Polarimetric L-band Multi-beam Radiometer (PLMR) observations collected during Heihe Watershed Allied Telemetry Experiment Research, and the ground measurements are used to validate the results. The main findings contain: 1) Accurate SM estimates with RMSE and ubRMSE less than 0.03 $m^3/m^3$, exceeding that of 0.04 $m^3/m^3$ of Soil Moisture Active and Passive (SMAP) and Soil Moisture Ocean Salinity (SMOS) missions target accuracies, are obtained. 2) The uncertainty in SM inversion is quantified with the value less than 0.095 $m^3/m^3$. 3) The Bayesian PI improves the simulation performance of forward model via maximum likelihood estimation. 4) The 3P and 2P strategies result in different SM inversion uncertainties. Overall, the present Bayesian PI combining multi-channel L-band observations has obtained a desirable SM estimation and uncertainty quantification, which may offer an insight into the future SM inversion based on passive microwave remote sensing.





## 1. Introduction

Soil moisture (SM) is a key variable in the process of water and energy balance at the land surface (Zhao et al., 2015;Kim et al., 2015). It usually presents strong spatial and temporal variation, resulting in difficulties of its analysis from in situ measurements, especially over large areas (Chaouch et al., 2013). Instead, it is promising that SM at large scale can be retrieved from passive microwave remote sensing observations (Martens et al., 2015;Calvet et al., 2011).

SM retrieval algorithm based on passive microwave remote sensing has been witnessed a huge development and refinement over the last several decades. Initially, the retrieval algorithms are mainly based on empirical methods (Shi et al., 2001), which are relative simple and easily to implement. However, due to the comprehensive understanding on complicity of microwave emission process and SM estimation, the algorithms are continuously refined and improved, and several complex and advanced algorithms were proposed, e.g., iterative (Njoku et al., 2003) and look up table algorithms (Koike et al., 2004) for Advanced Microwave Scanning Radiometer-Earth observation mission (AMSR-E ), optimization of cost function (Wigneron et al., 2007) for Soil Moisture and Ocean Salinity (SMOS) mission (Kerr et al., 2001). It is recognized that the brightness temperature (TB) received by the microwave radiometers is related to SM, land surface roughness, vegetation parameters and so on. Thus, the strategy of simultaneously estimating several surface variables from passive microwave remote sensing observations is commonly applied. Generally, this strategy is usually based on the inversion of a microwave emission model or a radiative transfer model (RTM). A global optimization algorithm is usually unitized to search the solutions of the inverse problem by constructing a cost function. We call this algorithm the cost function algorithm. The RTMs represent the physical process of interaction between the microwave signal with the land surface parameters, and the optimization algorithm can most probably derive the globally optimal solution by numerous iterative computing. However, this traditional cost function algorithm uses the sole "optimal estimate" to represent the retrieved variable's values, while their confidential intervals and uncertainties cannot be derived from the approach. Considering this problem, the author previously presented a probabilistic inversion algorithm for SM estimation using active microwave remote sensing (Ma et al., 2016). The probabilistic inversion (PI) approach is able to derive the posterior probability distribution of SM and to quantify the uncertainty in SM inversion. Thus, one of the main purposes of the present research is to verify the feasibility of the PI on the use of passive microwave through its proper improvement.

Besides, most of the presently released SM products based on passive microwave remote sensing are derived from the space-borne sensors. These products possess a coarse spatial resolution which limits the application of the products in watershed or oasis scaled regions. The airborne microwave radiometers can overcome this limitation, providing moderate or high spatial resolution soil moisture estimates. In many large-scale SM calibration/validation experiments, the airborne radiometers are widely used to provide SM estimates. For instance, Polarimetric Scanning Radiometer (PSR) (Jackson et al., 2002), Electronically Steered Thinned Aperture Radiometet (ESTAR) (Jackson et al., 1999) were used in South Great Plains (SGP) experiments (Jackson and Hsu, 2001;Famiglietti et



al., 1999) and Soil Moisture Experiments (SMEX) (Jacobs et al., 2004;Sano et al., 2004;Jackson et al., 2008), and Polarimtric L-band Multi-beam Radiometer (PLMR) was used in National Airborne Field Experiment (NAFE) (Panciera et al., 2008;Merlin et al., 2008) and Soil Moisture Active and Passive Experiments (SMAPEx) (Panciera
et al., 2014). PSR provide C-and X-band radiometer observations which are sensitive to SM but also to vegetation. The PLMR can provide L-band TB which is less influenced by the vegetation (Escorihuela et al., 2010;Schwank et al., 2005). As well as, the PLMR sensor operates at dual-polarized and multi-angular configuration, enhancing the observation ability. During the Heihe Watershed Allied Telemetry Experiment Research (HiWATER) (Li et al., 2013), PLRM was applied in SM observation and several datasets were obtained. Besides, a large amount of ground
observations were simultaneously obtained (Wang et al., 2013), which provided rich dataset for the validation of remote sensing algorithm and products.

Furthermore, it is commonly recognized that the estimation of surface roughness is the most crucial issue in SM estimation based on microwave remote sensing (Martens et al., 2015;Saleh et al., 2009) because of its high sensitivity to the microwave observations (Li et al., 2015;Wang et al., 2016;Ma et al., 2017;Neelam and Mohanty,
2015). Generally, the roughness can be classified into two categories, the physically based roughness and effective roughness. Measuring the roughness costs huge work and time, and it usually cannot be directly used in surface emission modeling and soil parameter inversion. Thus, the effective roughness (Hr) is widely applied. The Hr cannot be measured directly, and several study parameterized it as the function of measured roughness at L-band (Tsan and Schmugge, 1987;Wigneron et al., 2001), but these parameterizations needs to the measured roughness.
Thus, we usually estimate it prior or simultaneously to the estimation of SM. Several researchers (Saleh et al., 2007;Panciera et al., 2009;Richard et al., 2009) found that the Hr can be estimated via the linear relations between it and SM. However, the SM is usually the retrieval objective which is unknown. Martens *et al* (Martens et al., 2015) proposed two approaches to estimate Hr. The first approach utilized the TB to estimate effective roughness and the second approach utilized TB and leaf area index (LAI) to estimate the effective roughness.
The present paper intends to utilize two estimation strategies to retrieval SM. The first is the simultaneous estimation of SM, Hr and vegetation water content (VWC). The second utilizes Martens (Martens et al., 2015) approaches to estimated Hr and then simultaneously estimates SM and VWC. The purpose of this doing is to test whether the prior estimating Hr can reduce the uncertainty in the inversion and improve the SM estimates under the Bayesian PI framework.
The overall objective of this paper is to comprehensively evaluate the feasibility of Bayesian PI based on the multi-channel PLMR observations over heterogeneous cropland and to provide an estimates of the uncertainty in the inversion. The specific objectives include 1) to quantify the uncertainty in SM inversion and to improve the inversion accuracy via PI, 2) to evaluate of the impacts of different retrieval strategies on inversion uncertainty and accuracy, and 3) to examine the feasibility of Hr estimation approach in two aspects, the reduction of uncertainty
and the improvement of the inversion accuracy.

The remainder of the paper is organized as follows. The data used are described in detail in Section 2. The



methodology is briefly introduced in Section 3. The results are presented in Section 4. A deep discussion is provided in Section 5, and a summary of the main conclusions is presented in Section 6.

## 2.    Data and processing

The study area used in this paper is located in the mid-stream of Heihe River Basin (N 37.5–43, E 97–102), which is the second largest inland river basin of China. The land surface is mainly covered by man-made oasis and Gobi desert. The HiWATER (Li et al., 2013) was conducted in 2012. This paper selects a small area (Fig. 1) of the experiment filed which located in the desert-oasis transition zone for SM inversion and validation. The data used in this study includes airborne PLMR data and ground in situ data. The former is used to conduct PI for SM estimation and the latter is used to validate the results.

Fig. 1. The locations of study area and SM in situ campaign.

### 2.1. PLMR data and processing

PLMR is an L-band (frequency=1.41 GHz) radiometer, operating at multi-angles ($\pm7^{o}$, $\pm21.5^{o}$, $\pm38.5^{o}$) and dual polarizations (v & h). Thus it can provide 6-channel observations. During HiWATER, PLMR is conducted in 9 flights to measure the TBs in the middle and upper HRB. Due to differences in flight region and height, the spatial resolutions ranged from 300-750 m. The details of the data can be referred to (Che et al., 2013;Yan et al., 2015).

Due to the effect by radio frequency interference, there exists abnormal point in the collected data. To process this problem, we delete the data that exceed the validity rang (>300 K or <180 K) according to the actual condition of the study area. Then, the point cloud data is converted the raster according its spatial resolution. In this paper the data observed on July 10, 2012 is utilized to conduct SM retrieval experiment because it is found that the data collected in the day possesses the highest quality. The spatial resolution of the selected PLMR data is around 750 m.

### 2.2. In situ data

To provide the validation data for the inversion results, we simultaneously conducted ground in situ campaign concurrent with the airborne experiment. In-situ measurements of SM was collected. Samples of volumetric SM representative of the first 5 cm of the soil was taken using 6 Stevens Water Hydra Probe (Walker, 2010). These in situ SM measurements were collected along 11 transects of 21 samples (160-m row spacing and 80-m spacing between each sample), with 2 replicates for each sample to capture the short range SM variability, within a 1.6 km*1.6km square across Gobi desert-woodland-agricultural fields. All these data can provide validation for approximate 11 PLMR pixels as shown in Fig. 1. Additionally, vegetation (maize) was randomly sampled in the area to obtain VWC. The measured VWC is used to provide the initial value and ranges to model inputs.



## 3.    Methodology

This paper utilizes a Bayesian PI approach to estimate the SM based on airborne PLMR over vegetated field. The approach has been described in detail in (Ma et al., 2016;Xu et al., 2006;Hararuk et al., 2014). The posterior distribution of SM is derived based on Bayesian estimator, and uncertainty in the inversion is represented by several

statistics of the posterior distribution of retrieval objectives, including uncertainty (defined as the scale parameter of a general extreme value (GEV) distribution), skewness and kurtosis coefficients; the SM optimal estimates is represented by the maximum likelihood estimation (MLE). The general framework of the Bayesian PI is similar to our previous work in (Ma et al., 2016). However, the different aspects of this paper from the previous one can be summarized into 1) different forward models, 2) different data and, 3) different inversion strategies and, 4)

comprehensive results evaluation metrics. This paper uses 6-channel PLMR observations and the data has been introduced in Section 2. Here, the forward model, retrieval strategies and results evaluation metrics are briefly described.

### 3.1.  The L-MEB model

The L-band Microwave Emission from the Biosphere (L-MEB) model (Wigneron et al., 2007) is applied in this paper to model the microwave emission from the land surface. It is a semi-empirical microwave emission model based on a simplified solution of the zero-order radiative transfer equation, which can be regarded as the simplification of τ-ω model (Mo et al., 1982) at L-band. The detailed description of the model can be referred to (Wigneron et al., 2007). This paper only introduces the inputs and outputs of model because it is important for the

surface parameters inversion. The model inputs include soil-related, vegetation-related, roughness-related and sensor configuration-related parameters. Table 1 lists all the inputs and their meaning. The outputs of the L-MEB contain v- and h-polarized TBs. The researches of parameter sensitivity analysis (Li et al., 2015;Wang et al., 2016;Ma et al., 2017;Neelam and Mohanty, 2015) demonstrates that SM, Hr and VWC and effective soil temperature ($T_{eff}$) are the most sensitive parameter for TB. Indeed, the $T_{eff}$ is function of SM and soil temperature

and it can be estimated via parameterization scheme proposed by (Wigneron et al., 2007). The sensitivity of Teff is associated to the SM and soil temperature (Ma et al., 2017).Thus, this paper focuses on the estimation of SM, Hr and VWC.

**Table 1. The list of inputs of L-MEB.**

### 3.2.  Retrieval of SM, Hr and VWC (3P)

The complete Bayesian PI process contains several key steps, to determine prior information of retrieval objectives (to determine the ranges and initial values of the retrieval objectives), to construct likelihood function, to derive posterior distribution and quantify uncertainty, to get optimal estimates from the posterior distribution.

The ranges and initial values of SM ($SM_0(m^3/m^3)$) and VWC ($VWC_0$ ($kg/m^2$)) are set according to



investigation during HiWATER and those of roughness parameter ($Hr(-), Hr_0$) are given according to the investigation in (Li et al., 2015). The simultaneous estimation of 3 parameters is called the 3P estimation strategy. All these retrieval objectives are assumed to uniformly distribute. The parameter ranges and initial values are listed as following.

$$
\begin{array}{ll}
0.05 \leq SM \leq 0.50 & SM_0 = 0.24 \\
0.1 \leq Hr \leq 1.50 & Hr_0 = 0.6 \\
0.02 \leq VWC \leq 6.00 & VWC_0 = 1.2
\end{array}
\tag{1}
$$

The other insensitive parameters for L-MEB model are set to constants according to our experiment investigation as shown in Table 1.

The construction of likelihood function is based a cost function which is minimizing the differences between the 6- channels PLMR observed TBs and corresponding TBs simulated by L-MEB. The uncertainty quantification and MLE is very similar to the doing in (Ma et al., 2016). Thus, the details of constructing cost function, iteration process and MLE can be referred to (Ma et al., 2016).

### 3.3. Retrieval of SM and VWC (2P)

This retrieval strategy firstly estimates Hr using *Martens* (Martens et al., 2015) proposed approaches as shown in Eqs. (2) & (3):

$$
H_{rp}(TB_p) = (a_{1p} + a_{2p}TB_p)^{a_{3p}}
\tag{2}
$$

$$
H_{rpe}(TB_p) = (a_{1p} + a_{2p}TB_p + a_{4p}LAI)^{a_{3p}}
\tag{3}
$$

where $H_{rp}$ and $H_{rpe}$ are Hr estimated using different methods; $a_{1p}$, $a_{2p}$, $a_{3p}$, $a_{4p}$ (p represents polarization, h, v, ) are empirical coefficients which are polarization-dependent; LAI is leaf area index which related to the VWC. For simplicity, we assume $H_{rh} = H_{rv}$ and use the values of $a_{1h}$, $a_{2h}$, $a_{3h}$ and $a_{4h}$ and average values of three incidence- angular TBs to calculate the Hr. The range of LAI is 3<=LAI<=4 at the experimental region, thus Eqs. (2) & (3) are:

$$
H_{rh}(TB_h) = \left(-2.06 + \frac{0.0111(TB_{h7}+TB_{h21.5}+TB_{h38.5})}{3}\right)^2
\tag{2'}
$$

$$
H_{rh}(TBE_h) = \left(-1.58 + \frac{0.0117(TB_{h7}+TB_{h21.5}+TB_{h38.5})}{3} - 0.18LAI\right)^2
\tag{3'}
$$

According to the investigation of LAI and VWC of the study area, we get the relation between LAI and VWC as Eq. (4):

$$
LAI = 1.67\,VWC + 1.4
\tag{4}
$$

Then the estimated Hr is used as one of inputs of L-MEB to estimate SM and VWC simultaneously. The estimation strategy of SM and VWC utilizing the Eq. (2') calculated Hr is called 2P_TB strategy and that utilizing the Eq. (3') calculated Hr is called 2P_TBE strategy.





### 3.4. Validation metrics

Different from the point-pixel comparison validation strategy, all the ground measurements falling into a certain PLMR pixel are averaged to get pixel scale ground reference value. This value is applied to compare with the PI estimated SM. The standard validation indices of SMAP SM products (Entekhabi et al., 2010) and relative error (RE), including RMSE, the mean bias, the unbiased RMSE (ubRMSE), and coefficient of determination ($R^2$) are used. The definitions of the validation indices as following:

$$RMSE = \sqrt{E[(\text{SM}_{est} - \text{SM}_{ref})^2]} \tag{5}$$

$$Bias = E[\text{SM}_{est}] - E[\text{SM}_{ref}] \tag{6}$$

$$ubRMSE = \sqrt{E[((\text{SM}_{est} - E[\text{SM}_{est}]) - (\text{SM}_{ref} - E[\text{SM}_{ref}]))^2]} \tag{7}$$

$$R = E[(\text{SM}_{est} - E[\text{SM}_{est}])(\text{SM}_{ref} - E[\text{SM}_{ref}])]/(\sigma_{est} \cdot \sigma_{ref}) \tag{8}$$

$$RE = E\left[\frac{abs(\text{SM}_{est} - \text{SM}_{ref})}{\text{SM}_{ref}}\right] * 100\% \tag{9}$$

in above equations: $\text{SM}_{est}$ and $\text{SM}_{ref}$ represent the estimated and ground observed SM values, respectively; $E[\cdot]$ and *abs* are the expectation and absolute value operators, respectively; $\sigma_{est}$ and $\sigma_{ref}$ are the standard deviations of the estimated and ground measured SM, respectively.

## 4. Results and validation

### 4.1. Posterior distribution of SM and uncertainty in the inversion

Fig. 2 shows the probability distribution of 3P retrieval objectives. The uncertainty, skewness and kurtosis coefficients of SM posterior distribution are 0.079 $m^3/m^3$, 0.715 and 2.788, respectively. This uncertainty index describes the dispersion of the SM posterior distribution. It does not represent the error in SM estimates but it may cause the error in SM estimates. The skewness and kurtosis coefficients quantitatively describe the shape of the posterior distribution. The shape of SM posterior distribution and the value of skewness coefficient shown that SM is positively skewed. The kurtosis coefficient is larger than that of uniform distribution but smaller than that of normal distribution. Thus, SM is distributed non-uniformly and non-normally, indicating that SM distribution is well constrained by the PI but the inversion exists uncertainty. The figure also shows that SM observation value and MLE are very closed, which means the MLE may well represent the SM estimates.

Fig. 2 The probability distribution of 3P retrieval objectives

Figs. 3 & 4 show the results of 2P strategies estimated retrieval objectives. For the comparability, we select the results of the same pixel (pixel 5) as result of 3P. It can be seen that the SM distributions exhibit larger uncertainty index and smaller kurtosis, which means that the SM estimations of 2P strategies exist larger uncertainty



than that of 3P estimation. This uncertainty may be caused by the estimation of Hr. However, the VWC distributions
of 2P strategies show smaller uncertainty and larger kurtosis. This is because the Hr estimations in 2P strategies are
related to VWC, which may enhances the sensitivity of VWC on TB.

Fig. 3 The probability distribution of 2P_TB retrieval objectives

Fig. 4 The probability distribution of 2P_TBE retrieval objectives

### 4.2. Optimal estimation and validation of SM

Based on the posterior distribution of SM, the MLE of GEV is applied to represent the optimal estimates of SM.
Figs. 5-7 show the comparison of retrieved and ground measured SM. The error metrics are also shown in the
figures. It can be observed that all the retrieval strategies get satisfactory performances with very small RE, RMSE
and ubRMSE and relative large $R^2$ values. These retrievals have exceeded the expected accuracies of 0.04 $m^3/m^3$ of
SMAP and SMOS missions. Comparatively, the 2P retrieval strategies result in more excellent results than that
results from 3P strategy. The 2P strategies slightly overestimate the SM with positive bias but the 3P strategy leads
to a slightly underestimation of SM.

Fig. 5 The comparison of 3P strategy estimated and ground measured SM.

Fig. 6 The comparison of 2P_TB strategy estimated and ground measured SM.

Fig. 7 The comparison of 2P_TBE strategy estimated and ground measured SM.

### 4.3. Evaluation and improvements of Simulated TB

The optimized soil and vegetation variables obtained via MLE are used to model the TB with L-MEB model, and
then the simulated TBs are compared against those observed by PLMR (Figs. 8 -10), Small RMSE and large $R^2$ are
observed, indicating that the simulated TBs are improved by this Bayesian PI. Due to the lack of available Hr and
sufficient VWC measurements, we cannot directly compare these results with those modeled from the originally
observed surface variables. However, Yan (Yan et al., 2015) conducted parameter calibration for L-MEB model
utilizing the same dataset, the findings of which can provide reference for our work. By comprising, we can find
that our results are similar even better than those obtained by Yan (Yan et al., 2015). Thus, the Bayesian PI presented
in this paper has ability of improving the performance of L-MEB.

Fig.8. The comparison of PLMR observed TB with those simulated by MLE variables under 3P strategy.
Fig.9. The comparison of PLMR observed TB with those simulated by MLE variables under 2P_TB strategy.



Fig.10. The comparison of PLMR observed TB with those simulated by MLE variables under 2P_TBE strategy.

## 5. Discussion

The paper performs an experiment on SM estimation based on a Bayesian PI and 6-channel airborne PLMR observations collected during HiWATER. The result is validated by the corresponding ground measurement. The inversion uncertainty is quantified and the inversion accuracy exceeds the expected accuracies of SMAP and SMOS missions. However, several key points in SM estimation based on microwave remote sensing is needed to further discuss and address.

The first import issue is the uncertainty quantification of SM inversion. As described in Section 3, the approach Bayesian PI presented in this paper is similar but not identical to our previous work in (Ma et al., 2016). Thus, new findings are obtained in this work associating the uncertainty quantification. In (Ma et al., 2016), we discussed the quantification of SM inversion based on X-band SAR and concluded that the use of X-band SAR on SM estimation over vegetated soils exhibited relative large uncertainty which was related to the uncertainty in scattering model and inversion algorithm at pixel scale, as well as to the limitation of X-band SAR. The inversion uncertainty of this paper is smaller than that of (Ma et al., 2016). This paper utilizes 6-channel observation to better constrain the posterior distribution of SM, making the inversion uncertainty decrease. Besides, the number of unknowns of this inversion is less than that in the previous inversion, especially this inversion utilizes one "effective" roughens parameter instead of two physically based roughness parameters in the previous study. Third, the L-band passive microwave observations are applied in this inversion. On the one hand, the passive microwave observation is more sensitive to SM(Ma et al., 2017) while the active microwave observation is more sensitive to roughness (Ma et al., 2015). On the other hand, it is commonly recognized that L-band is more suitable than X-band for SM estimation over vegetated soil (Rosenqvist et al., 2007;Chen et al., 2010;Panciera et al., 2014;Zhao et al., 2015).

Secondly, the issue of surface roughness estimation is one of the most crucial issues in SM estimation based on microwave remote sensing. The roughness parameter Hr used in L-MEB model is an "effective" parameter which cannot be measured directly. This paper utilizes two strategies (3P &2P) to estimate it. We have seen from the results that the 2P strategies result in an increasing uncertainty in SM inversion, but a higher accuracy in SM optimal estimation. These observations are not contradictory because we have stated that the uncertainty in SM inversion is not the error of the SM estimates. The uncertainty originates from the retrieval algorithm and data source (Ma et al., 2016). It represents constraining ability of algorithm and data to the posterior estimates of the retrieval objectives. In the 2P retrieval strategies, a variable with uncertainty, Hr, is estimated by TB or TB and LAI, and it gets a determined value. Thus, the uncertainty in Hr estimation is transformed to and re-assigned by the two other variables, SM and VWC. The error of the optimal estimates of SM in this paper is described by several error metrics. In addition to relating to the uncertainty of the inversion algorithm and data source, the estimation error is related to retrieval and validation strategies, validation dataset and number of the unknowns. In this paper, the sole



difference in 3P and 2P strategies is the difference of the number of the unknowns and retrieval strategy. The 2P
retrieval strategies result in a more desirable SM estimation accuracy, indicating that 1) the Hr estimation
approaches proposed by Martens (Martens et al., 2015) is feasible to SM estimation, and 2) the reducing of
unknowns can enhance the estimation accuracy of SM. Overall, we can conclude that both the two strategies
utilized in this paper result in desirable SM estimations.

The third issue is how many retrieval objectives should be simultaneously estimated and how many
observations should be used. The development of multi-frequency multi-angle and multi-polarized microwave
sensors (Oza et al., 2006;Notarnicola and Posa, 2003;Dong et al., 2014), provides very rich data for SM estimation
using multi-channels. PLMR provides 6-channel observations to estimate three or two unknowns. Theoretically,
three unrelated observations can provide sufficient data for estimating three unknowns. That is, we can
simultaneously estimate SM, Hr and VWC using any combination of 3-channel PLMR observations. What we
concern is that the differences in uncertainty reduction and accuracy improvement under the use of different
observations. In this paper, we don't conducted the related experiments. However, research (Li et al., 2014) showed
that the more use of observations, the result was more accurate. We have discussed that the 2P retrieval strategies
result in larger uncertainty in SM inversion but more accurate SM estimates. Besides, researchers utilized spectral
index derived from optical remote sensing data, e.g., NDVI, to estimate VWC and then simultaneously estimate
SM and Hr. The premise of these doings is the accurate estimation of Hr or VWC. And these doing is expected to
introduce a larger uncertainty in SM inversion and get a more accurate SM estimates.

Finally, similar research was conducted to estimate SM using the same PLMR data (Li et al., 2014). The
authors explored three retrieval strategies using 1P, 2P and 3P and demonstrated that the 3P using 6-channel strategy
led to the best results of SM estimation. The conclusion is consistent to that we find in this paper. However, several
improved points can be observed in the present paper. Firstly, the previous one utilized traditional point-estimation
approach which only got the 'optimal' estimates of SM, while the present Bayesian PI results in the posterior
distribution of SM and uncertainty estimation of the inversion。The inversion accuracy is better than that of the
previous research. Secondly, the present work utilizes pure PLMR data but the previous study combined PLMR
and MODIS data. Third, in 2P strategy the present work estimate Hr using TB, but the previous work set it as
constant, which no doubt led to larger uncertainty. In 3P strategy, the present work determine the variance of the
PLMR observation according actual data but the previous work artificially set its value.

## 6.  Conclusions

The Bayesian PI has been demonstrated its ability of improving accuracy and quantifying uncertainty in the SM
estimation based on active microwave remote sensing. Meanwhile, the L-band radiometry has been witnessed a
huge potential in monitoring and estimating soil moisture over vegetated soils. The present paper combines the
merit of Bayesian PI with multi-channel airborne PLMR observations to develop highly accurate SM estimates as
well as to quantify the uncertainty in SM inversion. Various retrieval strategies and several retrieval experiments





are conducted based on PLMR data collected on July 10, 2012 during HiWATER experiments. The uncertainty in
SM inversion is estimated and highly accurate SM estimates are obtained through validating by simultaneous
ground measurements. The main finding can be summarized as following: (1) The 6-channel L-band passive
microwave observations can result in smaller uncertainty in SM inversion and more accurate SM estimates than
the X-band SAR observations conducted in previous work. The uncertainties in SM inversion are 0.079, 0.092 and
0.094 $m^3/m^3$ for 3P, 2P_TB and 2P_TBE strategies, respectively. All of the SM estimation accuracies are with the
RMSE and ubRMSE less than 0.03 $m^3/m^3$, which have exceeded the 0.04 $m^3/m^3$ of SMAP and SMOS missions
target accuracies. (2) In addition to quantify uncertainty in SM inversion and improve SM estimation accuracy, the
Bayesian PI can improve the simulation performance of L-MEB model via MLE, hence decreasing the differences
between L-MEB simulation and PLMR observation. This function is similar to the calibration of model parameter.
(3) Both the 3P and 2P strategies derive satisfactory SM estimates as long as they obtain an accurate roughness
estimation. But different retrieval strategies may result in difference in SM inversion uncertainties. (4) The Martens
(Martens et al., 2015) proposed approach can be used to estimate roughness from TB directly. And the estimated
roughness can assist to improve the SM estimation. Overall, the present Bayesian PI combining multi-channel
PLMR observations has obtained a desirable SM estimation and uncertainty quantification, which may offer an
insight into the future SM inversion based on passive microwave remote sensing.

**Acknowledgement**

This study is jointly supported by the National Natural Science Foundation of China under Grant 91425303 and
Key Research Program of Frontier Sciences, Chinese Academy of Sciences, under Grant QYZDY-SSW-
DQC011.

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





**Table 1. The list of inputs of L-MEB.**

| Parameter | Value | Meaning |
|---|---|---|
| Freq | 1.41 | Frequency of radiometer(GHz) |
| Theta | 7, 21.5, 38.5 | Incidence angle (degrees) |
| SM | Variable | Soil moisture ($m^3/m^3$) |
| $T_{surf}$ | 300 | Surface soil temperature (K) |
| $T_{deep}$ | 285 | Deep soil temperature (K) |
| rob | 1.33 | Soil bulk density (g/cm3) |
| clay | 0.32 | clay content (0-1) |
| sand | 0.495 | sand content (0-1) |
| *$tt_p$ | 1 | Vegetation structure parameter (-) |
| *$\omega_p$ | 0 | Vegetation scattering albedo (-) |
| vwc | Variable | Vegetation water content ($kg/m^2$) |
| Tc | 300 | Vegetation temperature (K) |
| Hr | Variable | Roughness parameter (-) |
| *$Q_p$ | 0 | Polarization mixing parameter (-) |
| *$N_p$ | 0 | Roughness exponent (-) |

* Subscript p (=h, v) represents the polarizations.





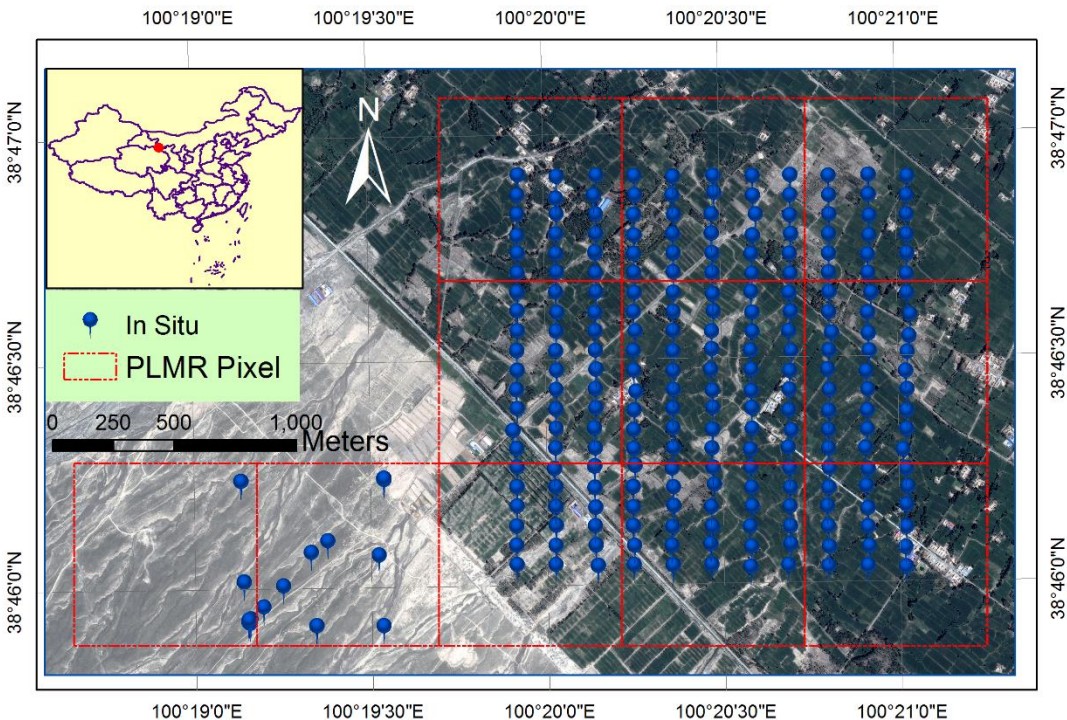

Fig. 1. The locations of study area and SM in situ campaign.





Fig. 2 The probability distribution of 3P retrieval objectives

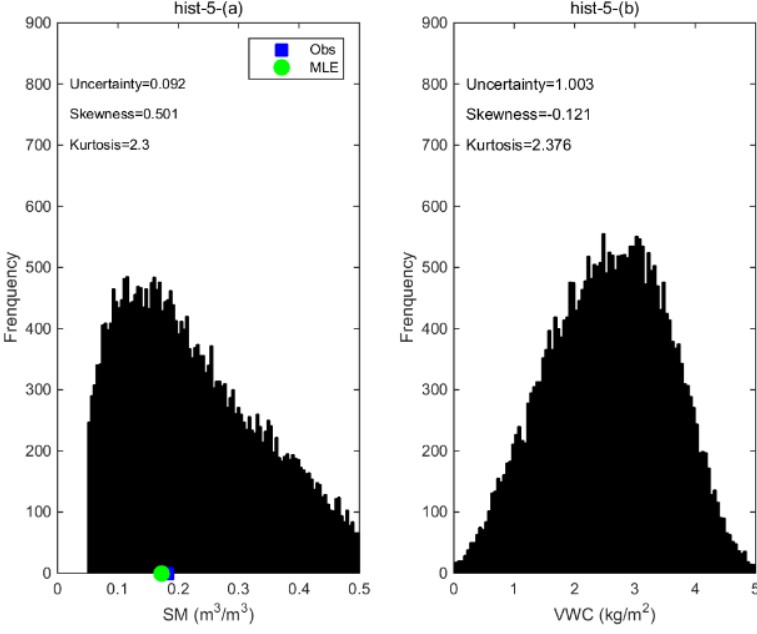

Fig. 3 The probability distribution of 2P_TB retrieval objectives


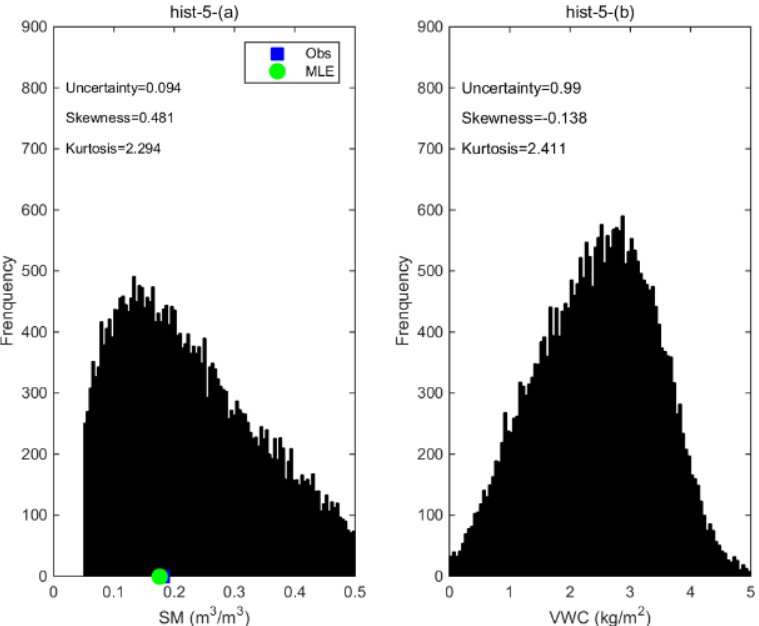

Fig. 4 The probability distribution of 2P_TBE retrieval objectives

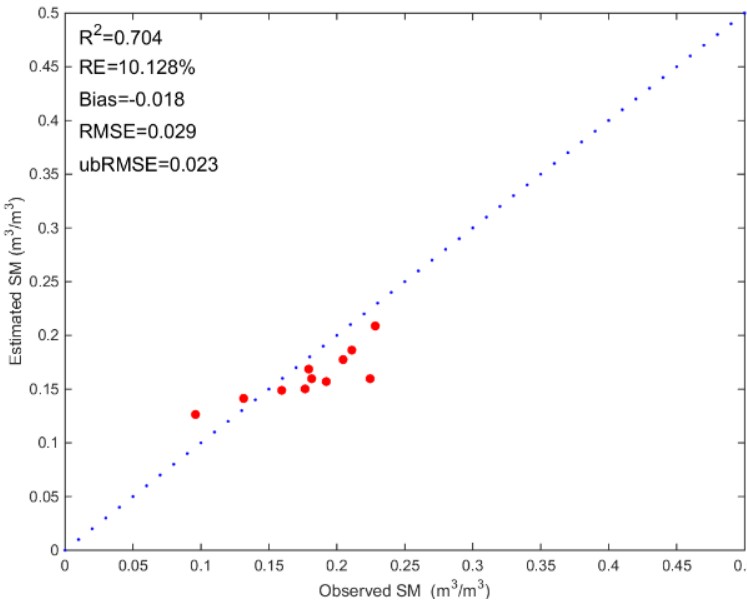

Fig. 5 The comparison of 3P strategy estimated and ground measured SM.




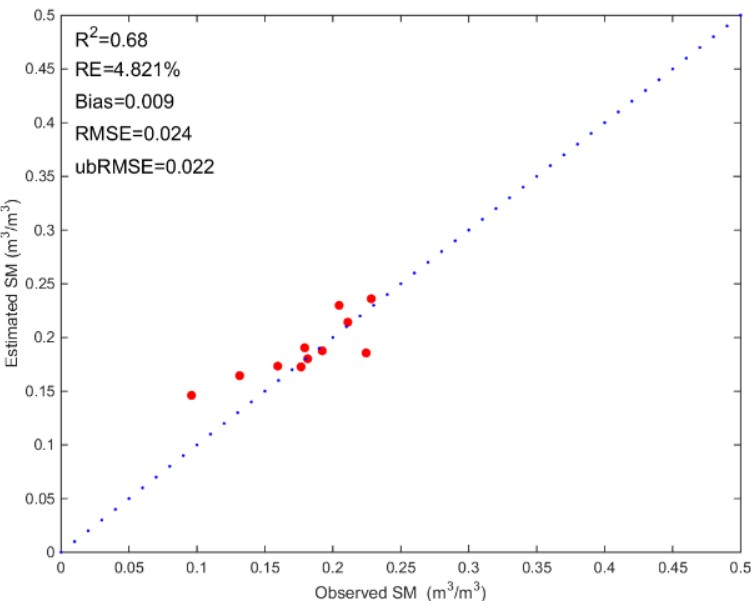

Fig. 6 The comparison of 2P_TB strategy estimated and ground measured SM.

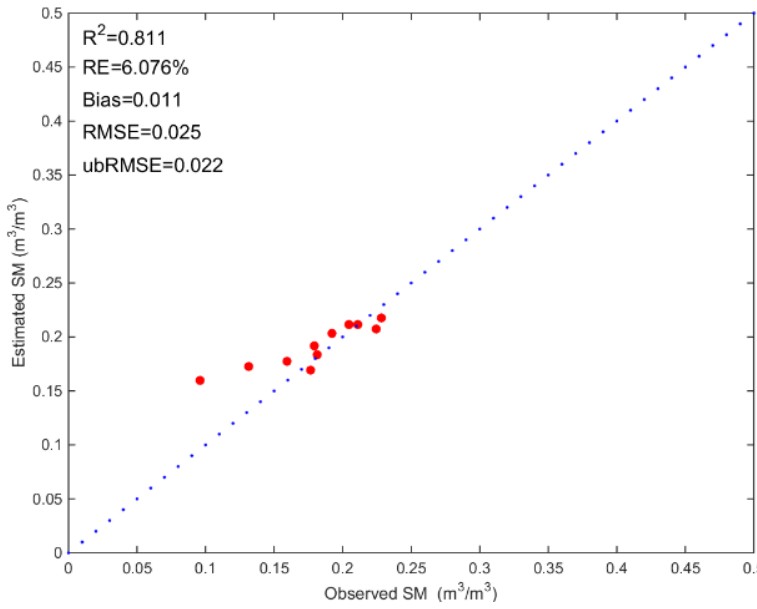

Fig. 7 The comparison of 2P_TBE strategy estimated and ground measured SM.





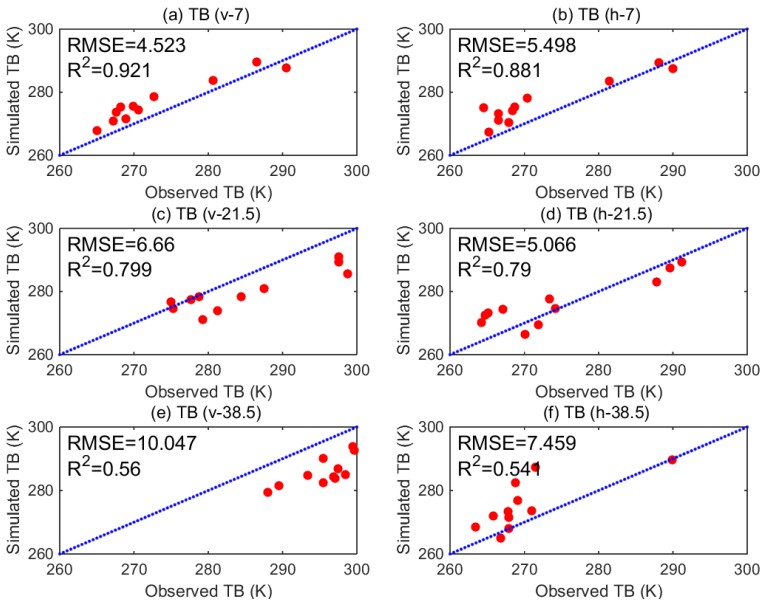

Fig.8. The comparison of PLMR observed TB with those simulated by MLE variables under 3P strategy.

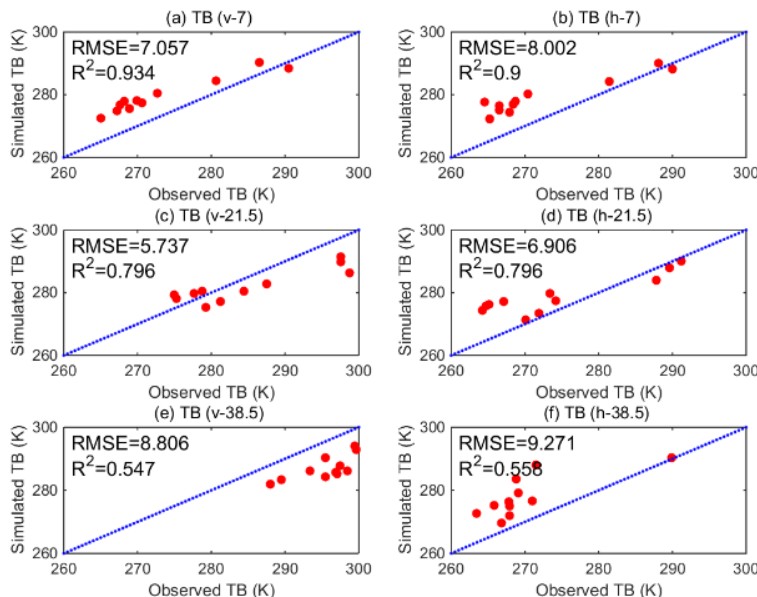

Fig.9. The comparison of PLMR observed TB with those simulated by MLE variables under 2P_TB strategy.




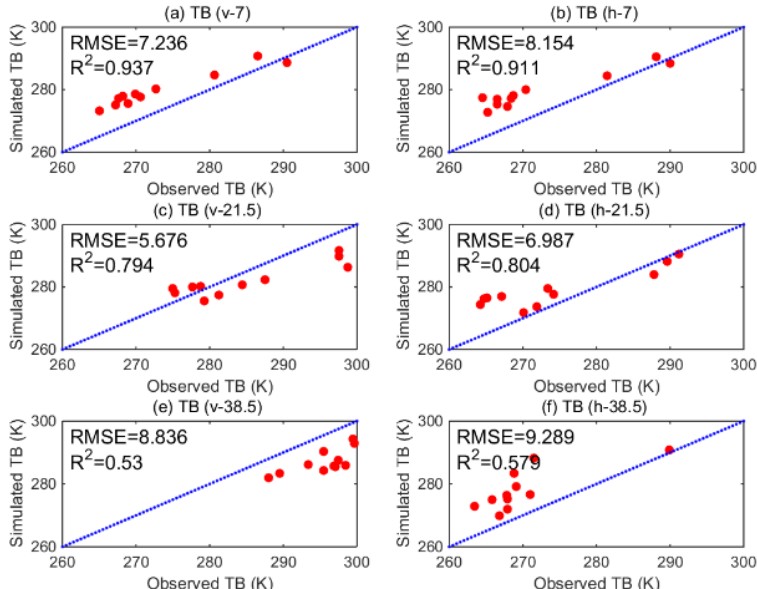

Fig.10. The comparison of PLMR observed TB with those simulated by MLE variables under 2P_TBE strategy.