# Peer review of "Soil Moisture Estimation Based on Probabilistic Inversion over Heterogeneous Vegetated Fields Using Airborne PLMR Brightness Temperature"

_Hydrology and Earth System Sciences, 2017_

## Referee Comment (RC1) · Anonymous Referee #1 · 3 Mar 2017

Review of the manuscript Soil Moisture Estimation Based on Probabilistic Inversion over Heterogeneous Vegetated Fields Using Airborne PLMR Brightness Temperature

by Ma et al.

The topic of this manuscript is up-to-date and very informative for HESS readers. The approach is sound, but there are some doubts that is has been implemented correctly. Several multi-parameter retrieval methods and applications have been published before. A discussion of these examples is missing but necessary to inform about the state-of-the-art in this respect and the related references need to be given. Similarly,

the intentions and results of previous PLMR applications should be discussed.

PLMR data processing should be described in more detail, a reference to previous publications is not sufficient to understand the approach. Hours and flight passes need to be given (just one day?). How did the authors handle different incidence angles? Was the data rasterized? How was soil surface temperature measured?

The L-MEB input data in Table 1 should be discussed a bit further. E.g., the use of fixed valued for soil texture and bulk density should be explained (e.g. by the small sized target region). Surface soil temperature and vegetation temperature (and to a certain extent deep soil temperature) cannot be fixed and leads to wrong estimates. It varies strongly during a flight and between dates. Additional temperature measurements are necessary.

Moreover, a brief introduction to the Bayesian approach and some equations should be given. In its current form it is not sufficient. Some more words why soil surface roughness is related to LAI (and VWC) should be provided. What is the relationship between VWC and vegetation opacity in this study (tau is the needed vegetation parameter, not VWC)?

Averaging brightness temperatures of three incidence angles to retrieve roughness parameters is physically not valid and leads to wrong estimates.

Please provide also the optimal parameters, i.e. a1p, a2p etc.

Additional language editing should remove small mistakes.

In total, due to several shortcomings in the presentation and doubts of correct method implementation I would reject this paper in its current form with request to re-submit after improvement.

Specific comments:

L. 15: L-band radiometers have . . . their. . .

L. 17: what are 'defects of the point-estimation algorithms'?

L. 24: not exceeding, but go below.

L. 26: introduce PI.

L. 46ff: give references.

L. 62: What are the characteristics of oasis environments and the challenges for SM retrieval?

L. 74: PLMR.

L. 80: What is the difference between the roughnesses?

L. 90: Retrieve.

L. 105ff: Some more information about the area of interest should be given.

L. 118: What is HRB?

L. 191ff: average TBs from different angles is not valid.

L. 278: important.

L. 286: roughness.

L. 301f: Please discuss the metrics.
* * *

---

## Referee Comment (RC2) · Anonymous Referee #2 · 8 Mar 2017

The topic and scientific goals of this contribution are certainly interesting for HESS readers. The overall methodology seems to be flawless, but due to the lack of documentation of the implementation and assumptions, questions arise. Without a clear description of the validity of the assumptions, reproducing the presented results is impossible. In addition, several typographical errors are present in this contribution, raising the question if a careful final reading was performed before submitting this report. Due to these typos, the paper does not read fluently enough to be able to focus on the scientific story.

A detailed description of the Bayesian PI methodology is missing for a paper that uses Soil Moisture Estimation based on Probabilistic Inversion... as a title. Include how the technique works and explain what the differences are with Ma et al. 2016.

With the 2P strategy, it is assumed that the roughness for horizontal and vertical polarizations are the same without any testing. Numerous studies have found that there is a difference between the roughness of both polarisations, so why is it valid to assume they are the same here? This is also dependent on the incidence angles.

The parameters determined by Martens et al. (2015) were calibrated for a specific region, it might be better to perform a calibration of your own. I would suggest to perform a calibration of your own, the VWC can be used directly to calculate the roughness without the necessity to first calculate the LAI based on the VWC if you determine your own regression.

It is stated that in previous work, larger uncertainties were introduced due to the use of a constant roughness (line 330), but there are studies that show that a constant roughness or even a roughness equal to zero can perform very well. For more information, have a look at A. A. Van de Griend, M. Owe, "Microwave vegetation optical depth and inverse modeling of soil emissivity using Nimbus/SMMR satellite observations", Meteorol. Atmos. Phys., vol. 54, pp. 225-239, 1994.

Some general remarks:

When an explicit citation in text is used, following structure should be used: name et al. (year). For example, Ma et al. (2016). Not "as stated by Ma (Ma et al., 2016)" or "as stated by Ma et al. (Ma et al., 2016)".

When multiple references are used, you should choose between ranking them chronologically or alphabetically, not randomly.

For figures, explain the legend also in the caption (what is visualised by the different colours). Also if both the x- and y-axis show the same variable (e.g. SM or TB), the

distance between the ticks on each axis should be the same. Use square plots in that case.

Be consistent in your reference list. For instance, not every article has a DOI, some DOI numbers have DOI in front of them, others do not.

Sometimes, the paper is written too informal. Extensive use of "we", "our" should be avoided. Also, previous study, this doing, its, etc. should be avoided.

A complete list of typos and minor comments can be found below. Based on the comments above and the list of minor comments below, I would reject this paper. I strongly advise to perform a screening of the text for additional typos and grammatical errors before resubmitting the revised paper.

Minor comments:

A brief method description is missing from the abstract. Include the model you are using as well as a short explanation of the strategies. At this moment, you introduce 2P and 3P strategies without explaining what it is, leaving the reader puzzled.

L41: complicity should be complexity

L42: . . .advanced algorithms were proposed. Some examples are iterative. . .

L46: Who recognised it? Multiple authors did, so refer

L49: unitized should be utilized

L51: . . .represent the physical interactions between the microwave signal and land surface parameters. The optimization. . .

L54: confidential intervals should be confidence intervals

L72: Also, the PLMR. . .

L74: PLRM should be PLMR

L75: . . .provided a rich. . .

L76: ...sensing algorithms. . .

L81: . . .costs time. . .

L82: modelling

L83: . . .as a function. . .

L84: . . .but these need parameterization to the. . .

L86: It is de Jeu, Richard (2009). Richard is his first name.

L86: . . .that Hr can. . .

L87: However, SM is. . .

L87: Explicit citation: Martens et al. (2015)

L89: . . .(LAI). (Remove last 5 words)

L90: . . . to retrieve SM.

L90: The first one is. . .

L91: The second strategy utilizes the technique proposed by Martens et al. (2015) to estimate Hr and then simultaneously estimate SM. . .

L93: . . . the prior estimated Hr. . .

L96: . . . an estimate. . .

L98: . . . to evaluate the. . .

L106: include ° sign for the geographical locations

L107: . . .the Gobi desert. . .

L109: . . .experiment field, which is located. . .

L117: . . .PLMR was conducted. . .

L117: What is v & h? Briefly explain what the difference is between vertical and horizontal polarisation.

L118: Explain HRB

L119: More details of the data can be found in Che et al (2013) and Yan et al. (2015).

L120: . . .effect of radio frequency interference, there exist abnormal points. . .

L121: Also, to process the problem of radio frequency interference, you use a validity range of 180-300K. Are these values chosen arbitrarily? Explain why exactly these values are used.

L121: . . .validity range. . .

L122: . . .is converted to a raster format according to its. . .

L123: The data of July 10, 2012 has the highest quality, but how is the quality checked?

L123: . . .because highest data quality is reached here.

L128: . . .conducted a ground in situ. . .

L129: In-situ should be in situ

L130: . . .soil were taken. . .

L131: 21 samples of SM were collected along. . .

L131: . . .(160m row spacing and 80m spacing between each sample),. . .

L133: . . .within a 1.6x1.6 km$^2$ squared area. . .

L134-135: Which procedure did you use to determine the VWC of maize? More detail is needed in this section.

L135: . . .ranges for the model input.

L137: . . .vegetated fields.

L138: . . .in detail by Ma et al (2016), Xu. . .

L141: . . .optimal estimates are represented. . .

L143: explicit citation

L143: However, this contribution can be distinguished from that of Ma et al. (2016) based on 1) the forward models, 2) data, 3) inversion strategies and 4) comprehensive. . .

L149: The L-MEB model is briefly described, but it is the main part of this contribution. Is has been used a lot, refer to some studies.

L153: A detailed description of L-MEB is given by Wigneron. . .

L153: What is a tau-w model, if it is mentioned in the text, an explanation is necessary.

L154: . . .outputs of the model. . .

L155: . . .soil-, vegetation-, roughness- and sensor. . .

L158: . . .SM, Hr, VWC and effective soil temperature are the most sensitive parameters. . .

L160: explicit citation

L160: Teff is not as visualised as a symbol

L161: a space after the point

L164: Table 1 contains all the parameters of L-MEB, without an explanation where the used values come from. Are these values from another study or are they calibrated? An answer is given at line 176, but this needs to be explained earlier

L166: Parts 3.2 and 3.3 have respectively 3P and 2P in their title, without an explanation. Briefly introduce these abbreviations before you start to describe the L-MEB

model.

L169: . . .posterior distribution, to quantify uncertainty and to get. . .

L170: SM0 and VWC0 has another font

L172: explicit citation

L178: . . .is based on a cost function which minimizes the. . .

L179-180: It is stated that the uncertainty quantification and MLE are very similar to the methods performed by Ma et al. (2016). However, what are the differences? Also, briefly explain how they work.

L180: explicit citation

L180: Ma et al. (2016) provide more details for constructing. . .

L184: . . .the approach proposed by Martens et al. (2015) is shown. . .

L189: . . .h,v). . .

L192-194: Refer to Martens et al. (2015) when the parameters of equation 2 and 3 are used in equations 2' and 3'.

L195: avoid the use of we

L199: . . .utilizing Eq. (2') to calculate Hr is called 2P_TB strategy and that Eq. (3') is called. . .

L204: . . .to get a pixel scale. . .

L221: . . .estimates, but. . .

L223: . . .and the skewness coefficient indicate that SM. . .

L226-227: It is stated that the SM observation value and MLE are similar, but if Figure 2 is checked, you can see there is a difference. The exact difference cannot be seen

because not enough xticks are present on the x-axis. However, are these values statistically similar? The range of SM is 0.05-0.5, so a deviation of 0.03 (my estimation of the difference between the SM observation value and MLE) is a difference of almost 10%.

L226: ...the inversion creates uncertainty.

L226: Fig. 2 also shows that...

L231: For comparability,...

L232: You mention that pixel 5 was used for the 3P strategy; however, this was not mentioned previously.

L232: ...(pixel 5) as performed for strategy 3P.

L233: ...strategies contain larger uncertainties...

L236: ...VWC to TB.

L 246: Reached accuracies are better than the target accuracies of SMOS and SMAP of 0.04 $m^3/m^3$. These are target accuracies for satellite platforms operating at a distance of 700 km, while PLMR operate at 300-750m. Comparing different methods when they are so different has to be performed with caution.

L249: ...a slight underestimation...

L257: ...simulated TB

L259: ...(Figs. 8-10). Small...

L261: ...those modelled using the...

L262: explicit citation

L263: avoid the use of "we"

L263: By comparing,...

L264: . . .are similar or even better. . .

L264: explicit citation.

L278: The first key issue. . .

L278: . . .Section 3, the Bayesian PI approach presented. . .

L279: It is stated that the method is different from that of Ma et al. (2016), but the difference is not explained.

L279: explicit citation

L280: explicit citation

L282: . . .in the scattering. . .

L286: inversion, mainly because this inversion. . .

L286: roughness

L287: . . .parameters as performed by Ma et al. (2016).

L289: The only references are to previous articles of one of the co-authors, while this is widely known and documented. Include some other references.

L298: It represents the constraining ability of the algorithm. . .

L306: explicit citation

L306: . . .feasible for SM. . .

L306: . . .2) reducing the number of. . .

L307: . . .both strategies. . .

L310: . . .multi-frequency, multi-angle. . .

L315: Not clear.

[Figure]

L316: . . .didn't conduct. . .

L317: . . .that if more observations are used, more accurate results can be obtained.

L318: Besides, a spectral index derived from optical remote sensing, e.g. NDVI, was used to estimate. . .

L320: Avoid the use of doing

L321: . . .accurate SM estimate.

L323: . . .using a 6-channel. . .

L325: Firstly, Li et al (2014) utilized

L327: Odd point is used to end the sentence

L328: Don't use previous, refer to the study

L328: . . .data, while Li et al (2014) combined

L328: Not entirely true. You use PLMR and in situ data of VWC to determine LAI

L329: Third, TB is used to estimate Hr in the present work, but Li et al. (2014) set. . .

L330: . . .undoubtedly. . .

L330: In 3P strategy, the variance of. . .. . ..is determined using actual data, but Li et al. (2014). . .

L334: . . .has demonstrated. . .

L335: . . .has shown a. . .

L340: . . .validating using simultaneous. . .

L341: . . .main findings. . .

L344: . . .accuracies (RMSE and ubRMSE) are less. . .

L348: . . .to the calibration function for model parameter.

L350: . . .result in differences. . .

L350: explicit citation

L400: title has no spaces

L437: it is de Jeu, R.

---

## Referee Comment (RC3) · Anonymous Referee #3 · 14 Mar 2017

This paper developed and tested the new algorithm of retrieving surface soil moisutre from passive microwave observations. Their Bayesian probabilistic inversion can quantify the uncertainties in their retrievals, which is the significant advantage of their proposed method compared with the previous algorithms.

I think this is an interesting paper. This paper is informative, and suitable to HESS. The method proposed in this paper provided the new contribution to the published knowledge.

However, I believe there are things to do described below to reach the full potential of

[Figure]

this paper. I recommend the editor to accept this paper after major revisions.

Major Points:

L180: Since Ma et al. [2016] has not been published yet, I recommend the authors to describe the short summary and important equations of their method in this paper. Please explicitly describe the cost function equation to be minimized. I guess that the authors used the Malcov Chain Monte Carlo (MCMC)-like sampling. Results of these interferences depend on the hyper-parameters of their probabilistic inversion approach. Please clarify their values and their sensitivities. This point is very important to interpret the author's results and make this paper solid.

L190-194: How did the authors get these values of empirical coefficients in equations (2) and (3)? Are they from Martens et al?

L197: Could the authors provide any references of equation (4)? How was this empirical relationship between LAI and VWC obtained? The authors mentioned that there are no VWC in-situ observations so that it might not be straightforward to obtain this relationship and the authors should explicitly explain how to get it.

L219: I believe that Figure 2, 3, and 4 show the retrieval result at a single grid point. Although these figures demonstrate how the author's algorithm works well, they do not comprehensively and quantitatively evaluate their results. In what conditions do the authors have a large uncertainty in their retrievals? Is estimated soil moisture highly uncertain in the case of high VWC? Please provide site-by-site comparison of the estimated uncertainties. I believe that the potential readers may be interested in this point because the uniqueness of the author's proposed method is the uncertainty quantification.

L219: Please explicitly explain how to calculate uncertainty, skewness, and kurtosis by showing equations. I hope this helps interpret the results.

L225-226: Why can the authors say "SM distribution is well constrained by the PI"? How

can the authors confirm that their retrievals are "well constrained?" Please explain this point more.

L226-227: Up to this point, results of only single grid are analyzed so that it cannot be proved that the MLE represent the SM estimates. As discussed above, please try to include all grid points in the analysis.

L250-255: Although I understand that the authors have no ground observations, I recommend the authors to include the comparison of estimated Hr and VWC between 2P and 3P algorithms. I guess that it helps interpret the difference of soil moisture retrieval performance shown in Figure 5, 6, and 7. In addition, please consider to include the uncertainty range in the estimated SM of Figure 5, 6, and 7. Again, the uniqueness of this paper is the uncertainty quantification so that the authors need to make more efforts on analyzing the estimated uncertainties.

L260: Why can the authors say "the simulated TBs are improved"?. When one can say "improved", one might compared the performance of their model with that of another model. Please consider to modify the expression of this point.

L264: Please explicitly describe the performance scores of Yan et al. [2015].

Minor Points:

L26: The authors have not mentioned that PI is the abbreviation of probabilistic inversion up to this point. Please write "Probabilistic Inversion (PI)".

L41: easily –> easy

L84: needs to the measured –> need to measure

L86: I recommend the authors to cite Wang et al. [2015] and Sawada et al. [2016] here. They also proposed the algorithm to objectively estimate the roughness parameters in radiative transfer models.

L121: rang –> range

L227: closed –> close

L264 our results are similar "OR" even better than. . ..

L280: Although Ma et al. [2016] is the authors' paper, I believe that here "we" should not be used. Please simply say "Ma et al [2016] discussed the quantification of. . ...".

L296-297: Maybe I missed something but where have you stated it?

L316: we don't conducted –> we did not conduct

L334: The Bayesian PI has been demonstrated –> The Bayesian PI has demonstrated

I think that there are many grammatical errors in the present form of the paper. I'm not capable of fixing all of them. I recommend the authors to carefully screen their manuscript to find and fix them.

References

S. Wang, J.-P. Wigneron, L.-M. Jiang, M. Parrens, X.-Y. Yu, A. Al-Yaari, Q.-Y. Ye, R. Fernandez-Moran, W. Ji, and Y. Kerr, "Global-Scale Evaluation of Roughness Effects on C-Band AMSR-E Observations", Remote Sens., vol. 7, pp. 5734-5757, doi: 10.3390/rs70505734, 2015.

Y. Sawada, H. Tsutsui, T. Koike, M. Rasmy, R. Seto, and H. Fujii, "A field verification of an algorithm for retrieving vegetation water content from passive microwave observations", IEEE Trans. Geosci. Remote Sensing, vol. 54, pp. 2082-2095, doi: 10.1109/TGRS.2015.2495365, 2016.

---

## Author Comment (AC1) · 9 Apr 2017

Dear Editor-in-Chief,

We are grateful for your patience and time in reviewing our manuscript "Soil Moisture Estimation Based on Probabilistic Inversion over Heterogeneous Vegetated Fields Using Airborne PLMR Brightness Temperature (hess-2017-34)". The comments of the three reviewers have been valuable in revising and improving our manuscript and serve as excellent guidance for our research. We did our best revising the manuscript, however, according to some comments we need re-conduct several numerical experiments

and rewrite certain contents. So we may spend some a long time to fully complete the revision. we noticed that the discussion will be closed by April 10. We are not sure whether we can catch this deadline, is it possible to have more days' extension?

Kind regards,

Chunfeng Ma, Xin Li, Shuguo Wang
* * *

---

## Author Comment (AC2) · 9 Apr 2017

Dear Referee,

Thank you for your valuable comments and suggestions for our manuscript. Your comments and suggestions are very helpful for improving our manuscript. We revised the manuscript item by item according to your suggestions. 1. We discussed the multi-parameter estimation in literatures as well as the use of PLMR data. 2. We described the PLMR data processing in detail. 3. We extended the description of L-MEB model and its inputs. 4. We added new a subsection (3.1) to introduce the Bayesian PI. 5.

[Figure]

As you suggested, the averaging of brightness temperatures at three incidence angles is not reasonable. This suggestion is undergoing a new scheme. We are conducting new numerical experiment. 6. We revised the editing and grammatical mistakes point-by-point according to your very careful and valuable suggestions.

Overall, we thank your comments very much. We have finished revising the major part of the manuscript. But the suggestion 5 is undergoing revision, we are afraid that we cannot upload the fully revised manuscript this time.

---

## Author Comment (AC3) · 9 Apr 2017

Dear Referee,

Thank you for your valuable comments and suggestions for our manuscript. Your comments and suggestions are very helpful for improving our manuscript. We revised the manuscript item by item according to your suggestions. 1. We added new a subsection (3.1) to introduce the Bayesian PI. 2. The dependences of roughness on polarizations and incidence angles are really a challenge issue. We are considering removing the 2P strategy or revising after testing. This issue is undergoing revision. 3. The empirical

parameters from Martens et al (2015) are directly used in our research site. This may result in errors. Although we made some measurements, it is not enough to fitting a robust relation between VWC and TB. So are considering removing the 2P strategy. 4. As you mentioned the impact of roughness on soil moisture estimation uncertainties, we reviewed the literature you suggested and revised the manuscript carefully. 5. The citation formats and figures of the whole manuscript have been revised carefully. 6. We are deeply touched by your so careful revision on the grammatical errors in the whole context. Your carefulness and spirit on scientific research is worth our study. We carefully revised the grammatical and editing mistakes point-by-point.

Overall, we thank your comments very much. We have finished revising the major part of the manuscript. The suggestion 2 is undergoing revision, we are afraid that we cannot upload the fully revised manuscript this time. But we will do our best to revise is as quickly as possible.

---

## Author Comment (AC4) · 9 Apr 2017

Dear Referee,

Thank you for your valuable comments and suggestions for our manuscript. Your comments and suggestions are very helpful for improving our manuscript. We revised the manuscript item by item according to your suggestions. 1. We added new a subsection (3.1) to introduce the Bayesian PI and the cost function, as well as, the results and discussion sections are extended and deepened. 2. Yes, the values of the empirical parameters. We are considering removing the 2P strategy because Martens et al uti-

**HESSD**

lized only incidence angle TB data, but we have 3 incidence angle TB data. 3. The Equation (4) is from our empirical fitting using the data from 2013 of our area. The relation was not published yet. In this edition, to make it general, we use a relation that has been published in PP.367 in: C. Matzler. Thermal Microwave Radiation: Applications for Remote Sensing. 4. Yes, we revised the figures and re-conducted experiments providing site-by-site comparison of the estimated uncertainties. 5. We added equations to describe the calculations of uncertainty, skewness and kurtosis. 6. Yes, we explained why "SM distribution is well constrained by the PI" in the discussion section. 7. Yes, we discussed the uncertainty, MLE on all grid points. 8. Thank you for the comment, we will compare the results from different strategies. This analysis is undergoing revision. 9. L260: Why can the authors say "the simulated TBs are improved"?. When one can say "improved", one might compared the performance of their model with that of another model. Please consider to modify the expression of this point. Yes, we revised the expression. 10. L264: Please explicitly describe the performance scores of Yan et al. [2015]. Yes, we described the scores of Yan et al (2015)

11. The grammatical errors in the whole context are carefully revised point-by-point.

Overall, we thank your comments very much. We have finished revising the major part of the manuscript. The suggestion 8 is undergoing revision, we are afraid that we cannot upload the fully revised manuscript this time. But we will do our best to revise is as quickly as possible.